# Susceptibility to positive versus negative emotional contagion: First evidence on their distinction using a balanced self-report measure

**Anton K. G. Marx** [1]*, **Anne C. Frenzel** [1], **Daniel Fiedler**[2], **Corinna Reck**[1]

**1** Department of Psychology, Ludwig-Maximilians-University (LMU) Munich, Munich, Germany,
**2** Department of Music Education, University of Erlangen-Nuremberg (FAU), Erlangen, Germany

* anton.marx@psy.lmu.de

## Abstract

Susceptibility to emotional contagion is defined as the disposition of how susceptible someone is to catch others' emotions and it has long been studied in research on mental health, well-being, and social interaction. Given that existing self-report measures of susceptibility to emotional contagion have focused almost exclusively on negative emotions, we developed a self-report measure to assess the susceptibility to emotional contagion of both positive and negative emotions (2 scales). In two studies, we examined their factor structure, validity, and reliability using exploratory factor analysis (Study 1, $N = 257$), confirmatory factor analysis (Study 2, $N = 247$) and correlations. Our results confirmed the two-factor structure and demonstrated good internal consistencies. Regarding external validity, our scales showed diverging correlational patterns: While susceptibility to negative emotional contagion was linked to mental health problems and negative emotions, susceptibility to positive emotional contagion was linked to interpersonal functioning and prosocial tendencies. In conclusion, our scales appear to be internally/externally valid and a promising tool for future research.

## Introduction

In a medical context, contagion is defined as transmission of diseases from one individual to another [1]. Accordingly, the term emotional contagion, in a psychological context, is used to describe the transmission of emotions between two or more individuals [2–9]. In addition to this interpersonal process, a trait-like disposition of how susceptible a person is to catch others' emotions has been proposed [5,6]: An individual's susceptibility to emotional contagion (SEC). In recent years, scientific interest in SEC has been growing; and previous studies have repeatedly linked an individual's SEC to increased negative emotions and mental health problems using self-report data [10,11]. However, the majority of items in existing self-report scales on SEC refer almost exclusively to the contagion of negative emotions. Thus, previous findings on SEC seem to be biased towards the contagion of negative emotions, and the SEC of positive

**Data Availability Statement:** All data and code necessary to reproduce the findings presented in this work are available in an OSF repository (see https://osf.io/jfxk4/) together with additional

supplemental material and Study 2 has been preregistered prior to conducting any analyses (see https://aspredicted.org/nk8zx.pdf).

**Funding:** The author(s) received no specific funding for this work.

**Competing interests:** The authors have declared that no competing interests exist.

emotions has been heavily neglected as of yet. To tackle this issue, the present contribution had two important goals. First, we sought to create and validate a balanced self-report measure to assess an individual's SEC of both positive and negative emotions separately. Second, we aimed to investigate the relations of positive vs. negative SEC to different variables of emotional experiences, interpersonal functioning, and mental health.

### Emotional contagion as an interpersonal process

The idea of emotional contagion has been around since the 18th century [12,13], and it was introduced to scientific fields related to psychology at least 100 years ago [14–17]. Since then, various terms have been used to describe this process, including contagion, transmission, transfer, crossover, resonance, or mirroring [3,7,18–23], and different underlying mechanisms have been proposed [4]. While previous studies have often been unclear and inconsistent in their conceptualizations and definitions [24], emotional contagion can be defined—in agreement with several theorists—as a basic interpersonal process that comprises the mostly automatic and unintentional transmission of emotions from an individual to one or more other individuals resulting in a more or less congruent emotional experience [2,5,7,8].

In that sense, it can be delineated clearly from the much broader concept of empathy, which itself has recently been discussed in recent reviews as a rather vague and manyfold umbrella term comprising a range of more precisely defined lower-level concepts (see [24–26], for critical discussions of empathy definitions). In line with these reviews, empathy definitions typically encompass understanding others' emotions and taking their perspective (i.e., cognitive empathy), sharing someone else's emotions and feeling prosocial concern, compassion, or sympathy for others (i.e., affective empathy), and sometimes even the behavioral tendencies and reactions resulting thereof as well as accurately recognizing others' emotional states [24–26]. However, this rather diverse collection of processes and capacities has been criticized and discussed as a threat to clear and consistent conceptualizations that are needed for scientific advancement and, instead, it has been suggested to bypass the term empathy in favor of more precise definitions of lower-level constructs, such as emotional contagion (e.g., [24,25].

Thus, in contrast to empathy, emotional contagion represents a purely affective response to other individuals' emotional experiences (instead of cognitive role-/perspective-taking), it is based on unintentional, automatic, and mostly unconscious processes (instead of deliberately imagining being in another person's situation), and it does not necessarily involve any behavioral reactions towards others (instead of comforting or helping another person). Most critically, though, emotional contagion does not require to distinguish between one's own emotions and the other person's emotions (instead of an at least minimum level of self-other distinction or awareness of another individual's situation that is postulated as a defining feature of empathy) [24–30]. In other words, emotional contagion means that another individual's emotions become my own emotions, while empathy means understanding and responding to another individual's emotional experiences while distinguishing between my response and the other person's emotions or situation.

In previous research, emotional contagion has been investigated within the context of social interaction in different applied fields, such as educational research [20,21,31,32] and clinical psychology [33–41]. In these studies, empirical evidence has been found for the contagious transmission of both positive [20,21,42] and negative emotions [18,32].

### Susceptibility to emotional contagion as a personality trait

Above and beyond being a general interpersonal phenomenon, emotional contagion has also been considered from a personality difference perspective. Based on the observation that some

individuals seem to catch others' emotions quite easily and other individuals seem to be relatively immune against such contagion, it has been speculated that people differ in how prone or susceptible they are to catching others' emotions. Hence, SEC has been proposed as a trait-like disposition. As such, an individual's SEC is thought to reflect how susceptible an individual is to catching others' emotions. More specifically, it is defined as an individual's tendency or proneness to automatically and unintentionally catch other individuals' emotions through the process of emotional contagion [5–8,43]. In that sense, SEC can be delineated from the broader concept of empathy, which commonly encompasses understanding others' emotions and taking their perspective (i.e., cognitive empathy), sharing someone else's emotions while maintaining self-other distinction (i.e., affective empathy), and even feeling compassion and sympathy for someone else (for critical reviews and discussions of empathy definitions, see [24–26]).

Methodologically, previous research relied on self-report measures to assess an individual's SEC. Following an extensive and systematic literature review (see supporting information S1 Appendix), we identified seven published scales that explicitly addressed SEC (e.g., the Emotional Contagion Scale [44] and the Emotional Empathic Tendency Scale [7]) and 21 scales focusing on different forms of empathy that contained items resembling emotional contagion. Of the seven scales addressing SEC explicitly, four targeted adults ([7,44–46]), one targeted adults and adolescents ([47]), one specifically targeted parents [48], and one targeted children [49], respectively. All of the seven instruments have been constructed based on the assumption that the personal tendency to catch other individuals' emotions through the process of emotional contagion is a global and unidimensional construct irrespective of the valence of the transmitted emotion.

Within all scales containing items on an individual's SEC, we identified 132 items that focused directly on being susceptible to emotional contagion in their wording. Conspicuously, most of these items focus on negative emotions (54%) as compared to items focusing on positive emotions (26%) or items with no explicit emotional valence (21%). Even more so, within the seven scales addressing SEC explicitly, we identified 50 items focusing directly on SEC, of which 60% addressed negative emotions, 24% addressed positive emotions, and 16% did not make the emotional valence explicit. Taken together, existing scales—and especially the ones explicitly addressing SEC which represent the basis of most previous literature on SEC—seem to be biased towards an individual's tendency to catch negative emotions and there seems to be a lack of more balanced measures to assess both the SEC of positive and negative emotions.

On a more substantial level, previous studies on individuals' SEC have largely focused on different socially interactive contexts, for example, among health care practitioners [10,50,51], social workers [11,52], or salespersons [53]. In these studies, self-reported SEC was linked to an increased experience of negative emotions and to various indicators of mental health problems (e.g., emotional exhaustion, depressive symptoms, or professional impairment). However, given that typically in prior research, composite global scores for SEC were used, it is unclear whether it is the susceptibility to negative or positive emotions, or both, which underlies the associations with these indicators of health problems and other aversive experiences. Due to the overemphasis on negative emotions in the items of the existing scales, we postulate that previous research on individuals' SEC is biased towards negative emotions. Research explicitly teasing apart the SEC of positive versus negative emotions is called for. To this end, a self-report measure to separately assess the susceptibility to catching positive and negative emotions in a balanced way is needed. While this separation may seem to be a solely methodological desideratum, it is also highly conceptually reasonable, as we set out below.

## Theoretical framework and empirical support for positive versus negative SEC

Originally, SEC has been conceptualized as a unidimensional and global construct. In the present work, we argue for a more complex perspective on an individual's SEC. Theoretically, our reasoning is rooted in Watson and Tellegen's [54] consensual model of positive and negative affect. In contrast to emotion theorists who argued for a specific set of discrete emotions [55–58], Watson and Tellegen proposed a more parsimonious model based on the reanalysis of several datasets with positive and negative affect as two general and relatively independent dimensions of emotional experiences that can be subdivided into specific discrete emotions reflecting the valence of the dimensions, such as happiness or anxiety [54,59–61].

Since Watson and Tellegen's proposal in 1985 [54], a range of empirical studies have provided evidence for this assumption. For example, in a study investigating college students' affect over several class sessions and how it is altered by the feedback on their performance, positive and negative affect were found to be unrelated in each measurement. Moreover, while success feedback increased positive affect, it did not impact students' negative affect, and vice versa [62]. In addition, several studies investigated certain context factors influencing the relationship between positive and negative affect. For instance, in a study examining the influence of culture and gender on the relationship between positive and negative affect, different correlational patterns were found for the subgroups of American men (weak negative correlation between positive and negative emotions) and women (strong negative correlation) versus Chinese men (weak positive correlation) and women (strong positive correlation), respectively [63]. Additionally, previous studies examining the influence of experienced stress on this association found that greater levels of stress lead to a stronger negative correlation between positive and negative affect [64,65]. In studies investigating the relation between positive and negative affect for both state and trait measures, the two dimensions were found to be unrelated at the trait level but showed a significant small negative correlation when considering a shorter time interval [66,67]. Taken together, the existing evidence suggests that people who frequently and intensively experience positive emotions do not systematically experience elevated or decreased levels of negative emotions. As such, the proposal of positive and negative affect as two separable and -more or less- independent dimensions provides a plausible and promising theoretical framework for our reasoning.

In line with this framework, we propose two distinct dimensions of SEC: The SEC of positive emotions (Positive SEC) and the SEC of negative emotions (Negative SEC). These dimensions reflect the valence of the emotions transmitted through the process of emotional contagion. In line with Watson and Tellegen's model, we do not necessarily expect an antagonistic relation between Positive and Negative SEC (i.e., a higher Positive SEC does not necessarily lead to a lower Negative SEC). While we do not rule out any covariation between the two dimensions, we do not propose that there is a strong higher-order personal tendency to catch emotions of either valence from other people (resulting in a strong positive correlation between Positive and Negative SEC and explaining most of the variance). Instead, we expect Positive SEC and Negative SEC to be largely independent from one another and linked differentially to variables in several important domains of human functioning, namely emotional experiences, social interaction, and individuals' mental and physical health.

Empirical support for a distinction of positive versus negative SEC comes from more recent studies in empathy research. For example, Light and colleagues [68] investigated an explicitly positive form of empathy using a newly developed self-report measure. They found that Empathic Happiness, an emotion-specific form of positive empathy (see also [69]), was negatively related to the experience of anhedonia and depressive symptoms [68]. Further,

Lundkvist [70] examined the factor structure of a Swedish adaptation of Doherty's [44] proposed initially unidimensional Emotional Contagion Scale (ECS) and reported that the items were grouped into two factors—SEC of positive and negative emotions. Similarly, Murphy and colleagues [22] investigated the validity of the Empathy Index [71], a self-report measure comprising the subscales Behavioral Contagion and Empathy, of which Jordan and colleagues [71] had explicitly claimed to represent a general form of being susceptible to the transmission of emotions. In their study, Murphy and colleagues [22] conducted factor analyses that revealed three different distinguishable factors, which they labeled Positive/Neutral Contagion (3 items, but only 2 refer to positive emotions; example item: "If I see a video of a baby smiling, I find myself smiling"), Distress Contagion (4 items; example item: "If I see someone fidgeting, I'll start feeling anxious too"), and Physical Mimicry, representing a purely motoric reaction to other individuals' behaviors (3 items; example item: "I catch myself crossing my arms or legs just like the person I'm talking to"). Subsequent correlational analyses revealed that Positive/Neutral Contagion was positively associated with sympathetic caring for others, extraversion, agreeableness, and openness, but negatively associated with attachment avoidance, emotional detachment, and various interpersonal aspects of personality disorders (e.g., meanness). Distress Contagion, on the other hand, was not associated with sympathetic caring, extraversion, agreeableness, and openness, but positively associated with anxiety, depressive symptoms, emotional distress, negative affect, attachment anxiety, and features of personality disorders (e.g., disinhibition). In a clinical context, Trautmann and colleagues [72] examined if an individual's tendency to vicariously experience others' positive versus negative emotions influences an individual's stress response when being exposed to a neutral film versus a film depicting a traumatic event. In this study, SEC was assessed using Lundkvist's [70] two-factor conceptualization of Doherty's ECS [44]. Trautmann and colleagues [72] found that being susceptible to negative emotions was positively related to the perceived stressfulness of the stimulus as well as increased anxiety and heart rate, whereas being susceptible to positive emotions was not related to these variables. However, this study focused solely on emotional responses to an extremely aversive and negatively valenced stimulus and it did not include any positively valenced stimuli or outcome variables. Nevertheless, their reported correlational pattern supports the distinction of positive versus negative SEC.

In conclusion, scattered empirical findings support our proposition that SEC should not be treated as a global and unidimensional construct. An individual's Positive SEC vs. Negative SEC appear to differ in their associations to relevant socio-emotional outcome variables (i.e., emotional experiences, interpersonal functioning, and well-being). Negative SEC has previously been associated with an increased experience of negative emotions and mental health problems [10,11,22,68,72]. Positive SEC has not shown similar or opposite relations in previous research, but has been associated with greater interpersonal functioning instead [22]. Taken together, there is both theoretical and empirical support for a distinct and independent conceptualization of Positive and Negative SEC.

## Study 1

Study 1 was part of a larger research project ([73]; see [74] for information on the overarching project) and designed to provide first evidence on the psychometric quality of our novel scales, their statistical separability, and their proposed differential relations with a relevant outcome variable, namely emotional exhaustion being one core component of burnout [75]. As elaborated above, we suggest that Negative SEC, but not Positive SEC, should be positively linked with this measure of psychological ill-being. In this study, we chose to sample from the population of secondary school teachers, given that teaching is not only highly emotionally

demanding and characterized by severe burnout risks, but also an extremely social and inter-personal profession [76,77].

## Materials and methods

### Sample and procedure

The sample of Study 1 consisted of $N$ = 257 teachers (70.4% female; no answer, 1.2%) from more than 40 schools in different German states (Bavaria, 81.3%; Baden-Wurttemberg, 11.3%; other, 6.6%; no answer, 0.8%). Their age ranged from 26 to 64 years ($M$ = 41.8; $SD$ = 10.9; no answer, 2.7%) and 96.9% had obtained the German "Abitur", a qualification level granted to students after usually 12 to 13 years after passing their final exams at the end of secondary edu-cation in Germany giving them access to academic higher education (no answer, 1.2%). They had an average teaching experience of 13.2 years ($SD$ = 10.3; no answer, 6.6%) and they worked at different types of secondary schools (13.6% lowest, 19.1% middle, 52.9% highest academic track), in primary schools (2.3%), in vocational schools (3.5%), or in more than one of these school types at the same time (6.2%; no answer, 1.6%). They taught a wide range of subjects (e.g., Math, German, English) and their average teaching hours per week ranged from 6 to 28 hours ($M$ = 20.7; $SD$ = 5.2; no answer, 5.1%). All participants were recruited voluntarily in 2017 through convenience sampling and direct contact with the schools or the teachers (i.e., schools and teachers were contacted via email, telephone, and written informational letters). Overall, more than 480 paper-pencil questionnaires were sent out, with a response rate of 53.2% resulting in our final sample size and exceeding response rates of previous paper-pencil studies in this specific population [78,79]. No incentives were given for participation and no identifiers that could link individual participants to their results were obtained. Thus, all analy-ses were conducted on anonymous data. All participants gave their informed consent and the study was conducted in accordance with the ethical standards of the APA and the Declaration of Helsinki. The research project received a formal waiver of ethical approval by the ethics review board of the first author's institution.

**Measures.**   Positive and Negative SEC—Scale Development and Item Selection: Within our two newly developed scales, we chose to formulate items with regard to specific discrete emotional experiences, such as being happy or stressed out. We aimed to create a balanced measure with items assessing the SEC of positive emotional experiences and of negative emo-tional experiences being equally represented. We were inspired by existing measures of SEC or affective empathy, including the ECS [44] and the Emotional Empathic Tendency Scale [7], and specific items within these measures served as the basis for newly created items. The selected items were all related to Positive SEC or Negative SEC in their content and wording and were either taken from published German translations or carefully transferred to the Ger-man language by means of translation-back-translation by two bilingual experts who reached a consent in a committee approach in case of alternative translations (see S1 Table). To elimi-nate existing deficits in published self-report measures of SEC, such as ambiguous items (e.g., "I am happy when I am with a cheerful group and sad when the others are gloom"; [46,80]) or competence-oriented items (e.g., "I am able to remain calm even though those around me worry"; [7]), some of the items have been reworded and new items have been created accord-ing to our two-fold conceptualization of SEC. In total, five items each were created to assess individuals' Positive SEC and Negative SEC. All items were presented in mixed order within the questionnaire and answered on a five-point Likert Scale (ranging from 1 = "strongly dis-agree" to 5 = "strongly agree").

Emotional Exhaustion: We used the German version of the emotional exhaustion subscale of the Maslach Burnout Inventory ([81,82]; 9 items; response scale 0 to 6; example item: "I feel

emotionally drained from my work"; Cronbach's alpha = .86; $M$ = 1.83; $SD$ = 0.96; Min/ Max = 0.11/4.56), which has been used frequently as a standalone measure of burnout in previous research [75].

**Statistical analyses.** First, we conducted exploratory factor analysis (EFA) to explore the factor structure of our newly developed scale, and potentially exclude items based on empirical and theoretical considerations; to result in a short and economic self-report measure. Second, we examined the psychometric properties of the scales and assessed their internal consistency. Given that McDonald's omega has the advantage of considering the strength of association between items and constructs as well as the item-specific errors [83], we used both Cronbach's alpha and McDonald's omega coefficients. Third, we calculated bivariate Pearson correlations to explore the relations between Positive and Negative SEC and with emotional exhaustion separately. Data processing and analysis were done in R and all data and code to reproduce the results of Study 1 are available in an OSF-repository together with additional supporting information (see https://osf.io/jfxk4/).

## Results and discussion

**Preliminary analyses.** All items aiming to assess an individual's SEC showed deviations from univariate normality as tested using the Shapiro-Wilk test, but no significant multivariate outliers were detected using the Mardia test [84]. While the items regarding Negative SEC appeared to be relatively normally distributed based on their skewness levels, those regarding Positive SEC appeared to be negatively skewed (see Table 1 for the items' psychometric properties). As tested using Little's MCAR test, missing data in all SEC items was completely at random and did not exceed 2% of all cases. These cases were listwise deleted resulting in $N$ = 252 cases for EFA.

**Exploratory factor analyses.** EFA was used to test our theoretical assumption of two separate dimensions underlying the idea of SEC. To investigate the adequacy of the collected data for factor analyses, we inspected the inter-item correlations and used the Kaiser-Meyer-Olkin (KMO) criterion and Bartlett's test of sphericity. Inter-item correlations ranged between .06 and .65, the KMO measure of sampling adequacy was .84, and Bartlett's test of sphericity was significant [$\chi 2(45)$ = 892.41, $p < .001$], collectively indicating good factorability of the data for further analyses ([85]). To determine the number of factors to retain, we first inspected the factors' eigenvalues revealing two factors with eigenvalues greater than one. Second, when

**Table 1. Psychometric properties of the final items of our SEC scale in Study 1.**

|  | | Min/Max | $M$ | $SD$ | Skewness | Kurtosis | Item total correlation if item deleted | α if item deleted |
|---|---|---|---|---|---|---|---|---|
| Positive SEC | | | | | | | | |
| | PSEC1 | 2/5 | 4.4 | 0.71 | -1.19 | 1.48 | .59 | .82 |
| | PSEC2 | 2/5 | 4.2 | 0.71 | -0.61 | 0.14 | .72 | .76 |
| | PSEC3 | 1/5 | 3.9 | 0.76 | -0.62 | 0.66 | .63 | .80 |
| | PSEC4 | 2/5 | 4.1 | 0.71 | -0.43 | -0.07 | .68 | .77 |
| Negative SEC | | | | | | | | |
| | NSEC1 | 1/5 | 2.58 | 0.93 | 0.21 | -0.45 | .56 | .70 |
| | NSEC2 | 1/5 | 2.62 | 0.94 | 0.01 | -0.73 | .45 | .76 |
| | NSEC3 | 1/5 | 2.97 | 1.11 | 0.05 | -0.77 | .56 | .71 |
| | NSEC4 | 1/5 | 2.88 | 0.92 | -0.08 | -0.59 | .68 | .64 |

*Note*. Following EFA, two items were excluded based on their factor loadings and/or cross-loadings resulting in 4 items per scale (see EFA results for detailed information). The response scale of all items ranged from 1–5 and, as tested using Shapiro-Wilks tests, the collected data showed deviations from normality for all items.

looking at the Scree Plot, we identified a break after two factors which were confirmed by parallel analysis. As this was in line with these results and our theoretical reasoning, we decided to retain two factors. As we would not exclude the existence of at least small-sized correlations between the two factors, we used an oblique factor rotation of the factors ("oblimin" method). Because the data violated the normality assumption, we used an ordinary least squares (OLS) method to find the minimum residual solution ("minres" method).

After conducting EFA, five items had significant factor loadings on Factor 1 and five items on Factor 2, respectively (ranging between .49 and .85). To create a parsimonious and economic yet still balanced measure of SEC, we excluded the two items with the weakest factor loadings and/or cross-loadings > .1 (see S2 Table for the factor loadings of all 10 items). Based on item content, Factor 1 was labeled "Positive SEC" and Factor 2 was labeled "Negative SEC" (see Table 2 for the factor loadings of the final 8 items in Study 1 and Study 2). The two factors showed an interfactor correlation of $r = .37$ and cumulatively accounted for 48% of the total variance (Factor 1 = 26%, Factor 2 = 22%). In sum, EFA revealed two separable yet positively related factors, and the items representing these two factors corresponded to our proposed theoretical conceptualization of Positive SEC and Negative SEC.

**Descriptive statistics.** Across all variables (incl. emotional exhaustion), missing values were found in 5.8% of the cases which were listwise deleted for subsequent descriptive and correlational analyses, which is thus based on $N = 242$ cases. Participants reported relatively high levels of item endorsement for Positive SEC ($M = 4.16$; $SD = .59$; Min/Max = 2/5) as compared to Negative SEC ($M = 2.76$; $SD = .75$; Min/Max = 1/4.5). However, standard deviations were sufficiently large to preclude ceiling or floor effects. Internal consistencies of the two scales were examined by calculating Cronbach's alpha and McDonald's omega coefficients showing acceptable to good reliabilities for Positive SEC ($\alpha = .83$; CI 95% [.80, .87] and $\omega = .83$; CI 95% [.80, .87]) as well as Negative SEC ($\alpha = .76$; CI 95% [.71, .81] and $\omega = .77$; CI 95% [.72, .81]), according to conventions [85].

**Correlational analyses.** Given the good internal consistencies of both SEC scales, we calculated mean scores for Positive SEC and Negative SEC. Bivariate Pearson Correlation revealed a significant positive relation between Positive SEC and Negative SEC ($r = .27$;

**Table 2. Factor loadings of the items of the positive and negative SEC scales in Study 1 and Study 2.**

| | Study 1 | | Study 2 | |
|---|---|---|---|---|
| | Factor 1 (Positive SEC) | Factor 2 (Negative SEC) | Factor 1 (Positive SEC) | Factor 2 (Negative SEC) |
| Positive SEC | | | | |
| PSEC1: It cheers me up to be around a jolly person. | .688 | | .657 | |
| PSEC2: It fills me with joy to be around happy people. | .832 | | .829 | |
| PSEC3: I let myself be infected by someone's enthusiasm. | .763 | | .663 | |
| PSEC4: I get cheerful when I am surrounded by cheerful people. | .705 | .170 | .800 | |
| Negative SEC | | | | |
| NSEC1: I get nervous when others around me are nervous. | | .652 | | .708 |
| NSEC2: I get angry when I am surrounded by enraged people. | | .514 | | .629 |
| NSEC3: I tense up when I hear people fighting. | | .682 | | .735 |
| NSEC4: I get stressed when I am around stressed people. | | .847 | | .876 |

*Note*. In Study 1, EFA was conducted to test the assumption of two separate dimensions of SEC. In Study 2, CFA was conducted to replicate the factor structure that was found in Study 1. For both studies, only factor loading >.1 are displayed.

$p < .001$; CI 95% [.15, .38]). Further, Negative SEC was significantly positively related to self-reported emotional exhaustion ($r = .27$; $p < .001$; CI 95% [.15, .38]), whereas Positive SEC and emotional exhaustion were uncorrelated ($r = -.02$; $p = .765$; CI 95% [-.14, .11]).

Taken together, Study 1 provided initial evidence for our theoretical assumption regarding two separable dimensions of SEC (Positive SEC vs. Negative SEC). Our findings suggest that our new balanced SEC measure represents a reliable instrument in terms of the scales' internal consistencies. Additionally, Positive and Negative SEC proved to differ in their relation to a theoretically meaningful external criterion. Negative SEC, but not Positive SEC, was linked with emotional exhaustion as a prototypical measure of psychological ill-being.

## Study 2

In Study 2, we had two primary goals: First, we aimed to corroborate and expand the evidence on our novel scales' validity and reliability. We expected to replicate the proposed two-factor structure in an independent and more generalized sample using confirmatory factor analysis (CFA), clearly identifying Positive SEC and Negative SEC as two distinct, yet related, factors, with the two-factor model's fit being superior to the single-factor model (internal validity). Second, regarding the scales' external validity, we aimed to investigate the associations of self-reported Positive SEC vs. Negative SEC with measures of emotional experiences, interpersonal functioning, health/well-being (criterion-oriented validity), and measures of cognitive and affective empathy components (divergent and convergent validity). Study 2 was preregistered prior to conducting any analyses (see https://aspredicted.org/nk8zx.pdf) and all data and code to reproduce the results of Study 2 are available in an OSF-repository (see https://osf.io/jfxk4/) together with additional supplemental material.

## Materials and methods

### Sample and procedure

The sample of Study 2 consisted of $N = 247$ participants (48.6% female) that were recruited in 2018 via Clickworker (www.clickworker.com), a German online recruiting service similar to Amazon MTurk (www.mturk.com). Their age ranged from 18 to 75 years ($M = 39.9$ years; $SD = 12.8$ years), 55.9% had obtained the German "Abitur", and 38.5% indicated to be married. All data was collected anonymously using SoSciSurvey, a German open-access provider for online-surveys (www.soscisurvey.de). Due to data privacy reasons, no information on the participants' place of residence or professional occupation was assessed. The participants received a small monetary incentive (3 €) for completing the survey. On average, they required $M = 21.2$ minutes to complete the questionnaire ($SD = 7.82$, Min = 6.2, Max = 59.7). While all participants who started to fill out the online questionnaire also completed it (completion rate = 100%), three participants were excluded prior to any further data processing, as preregistered, because they filled out the questionnaire unrealistically fast (i.e., in under five minutes). The acquired sample size ($N = 250$) was determined based on statistical power estimations with statistical power for correlational analyses of $1 - \beta > .99$ for medium and large effects (alpha level = 0.01). The study was conducted in accordance with the ethical standards of the APA and the Declaration of Helsinki and the research project was approved by the ethics review board of the first author's institution.

**Measures.** In addition to our two novel SEC scales, all participants completed the following self-report measures of (1) cognitive/affective empathy, (2) positive/negative emotionality and distress, (3) well-being and mental/physical health, and (4) social/interpersonal functioning (see Table 3 for sample items and detailed characteristics for each measure including example items and response scales).

**Table 3. Detailed characteristics and example items of all measures used in Study 2.**

| | Reference | N items | Example item | Response scale |
|---|---|---|---|---|
| Measures of Empathy | | | | |
| IRI Perspective Taking | [86] | 4 items | Before criticizing somebody, I try to imagine how I would feel if I were in their place. | 1 (never)– 5 (always) |
| IRI Fantasy | [86] | 4 items | After seeing a play or movie, I have felt as though I were one of the characters. | 1 (never)– 5 (always) |
| AMES Affective Empathy | [87] | 4 items | When people around me become nervous, I become nervous, too. | 1 (never)– 5 (always) |
| AMES Cognitive Empathy | [87] | 4 items | I can often understand how people are feeling even before they tell me. | 1 (never)– 5 (always) |
| Measures of Positive/Negative Emotionality and Distress | | | | |
| PANAS Positive Affect | [59] | 10 items | enthusiastic | 1 (not at all)– 5 (extremely) |
| PANAS Negative Affect | [59] | 10 items | nervous | 1 (not at all)– 5 (extremely) |
| IRI Personal Distress | [86] | 4 items | Being in a tense emotional situation scares me. | 1 (never)– 5 (always) |
| BFI2 Emotional Volatility | [88] | 4 items | I am someone who is temperamental, gets emotional easily. | 1 (strongly disagree)– 5 (strongly agree) |
| BFI2 Depressiveness | [88] | 4 items | I am someone who often feels sad. | 1 (strongly disagree)– 5 (strongly agree) |
| BFI2 Anxiety | [88] | 4 items | I am someone who can be tense. | 1 (strongly disagree)– 5 (strongly agree) |
| BFI2 Activity/Energy Level | [88] | 4 items | I am someone who can be tense. | 1 (strongly disagree)– 5 (strongly agree) |
| Measures of Well-being and Mental/ Physical Health | | | | |
| EDS Depressiveness | [89] | 10 items | In the past 7 days, I have been so unhappy that I have been crying. | 0–3 (different responses for each item) |
| GAD7 Anxiety | [89] | 7 items | Over the last two weeks, how often have you been feeling afraid as if something awful might happen? | 0 (not at all)– 3 (nearly every day) |
| PSS Stress | [90] | 10 items | In the last month, have you felt that you were unable to control the important things in your life? | 1 (never)–(very often) |
| CHIPS Physical Symptoms | [91] | 8 items | During the past three months, how much were you bothered by sleep problems? | 1 (not at all)– 5 (5 or more times a month) |
| SWLS Life Satisfaction | [92] | 5 items | The conditions of my life are excellent. | 1 (strongly disagree)– 7 (strongly agree) |
| Measures of Social/Interpersonal Functioning | | | | |
| AMES Sympathy | [87] | 4 items | I feel concerned for other people who are sick. | 1 (never)– 5 (always) |
| IRI Empathic Concern | [86] | 4 items | When I see someone being taken advantage of, I feel kind of protective towards them. | 1 (never)– 5 (always) |
| BFI2 Sociability | [93] | 4 items | I am someone who is outgoing, sociable. | 1 (strongly disagree)– 5 (strongly agree) |
| BFI2 Assertiveness | [93] | 4 items | I am someone who is dominant, acts as a leader. | 1 (strongly disagree)– 5 (strongly agree) |
| BFI2 Compassion | [93] | 4 items | I am someone who is helpful and unselfish with others. | 1 (strongly disagree)– 5 (strongly agree) |
| BFI2 Trust | [93] | 4 items | I am someone who assumes the best about people. | 1 (strongly disagree)– 5 (strongly agree) |
| BFI2 Respectfulness | [93] | 4 items | I am someone who is polite, courteous to others. | 1 (strongly disagree)– 5 (strongly agree) |
| Measure of Social Desirability | | | | |

(*Continued*)

**Table 3.** (Continued)

| | Reference | *N* items | Example item | Response scale |
|---|---|---|---|---|
| KSE-G Positive Qualities | [94] | 3 items | No matter who I'm talking to, I'm always a good listener. | 1 (strongly disagree)– 5 (strongly agree) |
| KSE-G Negative Qualities | [94] | 3 items | There have been occasions when I have taken advantage of someone. | 1 (strongly disagree)– 5 (strongly agree) |

Measures of Empathy: To assess different components of empathy, we used scales from the Interpersonal Reactivity Index (IRI; [86,95]) and the Adolescent Measure of Empathy and Sympathy (AMES; [87]), which has been validated in adult samples as well [96]. The IRI Perspective-Taking scale (Cronbach's alpha $\alpha$ = .76; McDonald's omega total $\omega$ = .81) measures an individual's tendency to adopt another's perspective to find out what another person might be thinking, thus, representing the cognitive empathy component. The IRI Fantasy scale ($\alpha$ = .75; $\omega$ = .81) measures an individual's tendency to imaginatively transpose oneself into the feelings and actions of fantasy characters. The AMES Affective Empathy scale ($\alpha$ = .81; $\omega$ = .84) aims to measure the tendency to vicariously experience another person's emotions which is quite similar to the concept of emotional contagion. The AMES Cognitive Empathy scale ($\alpha$ = .82; $\omega$ = .86) aims to measure an individual's capacity to cognitively understand another person's emotions.

Measures of Positive/Negative Emotionality and Distress: To assess different aspects of individuals' emotionality and distress, we used scales of the IRI and the Big-Five-Inventory 2 (BFI2; [93,97]) as well as the Positive and Negative Affect Schedule (PANAS; [59,98]). The IRI Personal Distress scale ($\alpha$ = .76; $\omega$ = .80) measures an individual's self-oriented tendency to experience distress and anxiety in highly emotional interpersonal situations, such as emergencies. Within the BFI2, the Big Five personality traits can be further subdivided in more differentiated facets. From the Negative Emotionality trait (formerly Neuroticism), we used the BFI2 Emotional Volatility ($\alpha$ = .75; $\omega$ = .80), BFI2 Depressiveness ($\alpha$ = .83; $\omega$ = .87), and BFI2 Anxiety ($\alpha$ = .69; $\omega$ = .77) facets and the BFI2 Activity/Energy Level ($\alpha$ = .73; $\omega$ = .79) facet from the Extraversion trait, respectively. The PANAS scales (Positive Affect and Negative Affect) assess an individual's general experience of positive ($\alpha$ = .88; $\omega$ = .91) and negative affect ($\alpha$ = .89; $\omega$ = .92) and consist of adjectives describing discrete emotional experiences that are either positive or negative in their valence (instruction: "How do you feel in general?").

Measures of Well-being and Mental/Physical Health: We used the Edinburgh Depression Scale (EDS; [88,99,100]) to assess current depressive symptoms ($\alpha$ = .87; $\omega$ = .90), the General Anxiety Disorder 7 questionnaire (GAD-7; [89,101]) to assess current symptoms of general anxiety ($\alpha$ = .88; $\omega$ = .91), and the Perceived Stress Scale (PSS-10; [90,102]) to assess individuals' current levels of stress ($\alpha$ = .87; $\omega$ = .90). To assess individuals' current physical symptoms ($\alpha$ = .80; $\omega$ = .86), we used a short version of the Cohen-Hoberman-Inventory of Physical Symptoms (CHIPS; [91,103]) and to assess individuals' satisfaction with their lives and current living conditions ($\alpha$ = .93; $\omega$ = .94), we used the Satisfaction with Life Scale (SWLS; [92]).

Measures of Social/Interpersonal Functioning: To assess different aspects of individuals' social and interpersonal functioning, we used scales of the IRI, the AMES, and specific facets of BFI2 personality traits. The AMES Sympathy subscale ($\alpha$ = .68; $\omega$ = .77) aims to measure an individual's tendency to feel concerned or sorrow for another person. The IRI Empathic Concern subscale ($\alpha$ = .69; $\omega$ = .71) aims to measure an individual's tendency to feel concern and sympathy towards others. From the personality trait extraversion, we used the BFI2 Sociability ($\alpha$ = .88; $\omega$ = .82) and BFI2 Assertiveness ($\alpha$ = .78; $\omega$ = .80) facets and the BFI2 Compassion ($\alpha$

= .64; ω = .71), BFI2 Trust (α = .61; ω = .73), and BFI2 Respectfulness (α = .73; ω = .80) facets from the personality trait Agreeableness, respectively.

Social Desirability: Given that self-reported health-related and clinical psychological variables have been reported to be associated with social desirability [104], we decided to assess the participants' social desirability response tendencies. To this end, we used a short German measure (KSE-G; [94]) of the tendency to either exaggerate one's positive qualities (α = .62; ω = .64) or to conceal negative qualities (α = .70; ω = .72)

**Statistical analyses.** Regarding the scales' internal validity, we conducted CFA using the package "lavaan" ([105], version 0.6.4) in R ([106], version 3.6.0). To evaluate model fit, we inspected a range of fit indices, including the comparative fit index (CFI), the Tucker Lewis Index (TLI; also called the non-normed fit index), the root-mean-square error of approximation (RMSEA), and the standardized root-mean-square residual (SRMR). In line with Hu and Bentler [107], model fit was recognized as acceptable with a CFI and TLI of .95 or higher, an RMSEA of .06 or lower, and an SRMR of .08 or lower [107,108]. Because the data violated the normality assumption, we used robust estimators of model fit (MLR) with robust (Huber-White) standard errors and a scaled test statistic that is (asymptotically) equal to the Yuan-Bentler test statistic [109]. We conducted model comparisons using the "anova" function ($\chi^2$ Difference Test with "satorra.bentler.2001" method) in the R package "lavaan".

Given that several variables were substantially related to self-reported social desirability, we calculated partial correlations of the two scales (Positive and Negative SEC) with these measures while controlling for social desirability (see S3 Table for bivariate correlations of all measures with social desirability and S4 Table for bivariate zero-order correlations of Positive SEC and Negative SEC with all measures without controlling for social desirability). While the distribution of almost all variables were judged as non-normal based on Shapiro-Wilk tests, we still consider the distribution of our variables sufficiently normal based on visual inspection of the collected data (all data visualizations are included in the analysis scripts in the supplement) and the respective skewness levels [85], especially in light of the generally assumed robustness of Pearson correlations against violations of the normality assumption (e.g., [110]; see S5 Table for supplemental non-parametric Spearman correlations of Positive and Negative SEC with all measures.). To account for multiple testing and the risk of alpha cumulation, we adjusted alpha significance levels using the Bonferroni correction for 23 correlational tests resulting in a new alpha significance level of $p < .002$ and calculated confidence intervals for all correlation coefficients [111]. As in Study 1, all data processing and statistical analyses were done in R, and reproducible scripts for all reported results were generated.

## Results

**Descriptive statistics and reliability.** Table 4 shows descriptive statistics for all Positive and Negative SEC items. Again, relatively strong item endorsement was found for Positive SEC (means > 3 on the five-point scale for all items) as compared to Negative SEC (means < 3 on the five-point scale for all items), yet with standard deviations again being sufficiently large to preclude ceiling or floor effects in both scales. Item total correlations (part-whole-corrected) ranged between $r = .59$ and .73 for Positive SEC and .57 and .75 for Negative SEC, indicating good item discrimination capacities and item homogeneity in both subscales [86]. All items of the SEC scales deviated from univariate normality as tested using the Shapiro-Wilk test. Still, no significant multivariate outliers were detected using the Mardia test [85]. Based on their skewness levels, the items regarding Negative SEC seem to be relatively normally distributed, whereas the items regarding Positive SEC appear to be negatively skewed. No missing values were found in the responses to the items of the SEC scale.

**Table 4. Descriptive statistics of all items of our positive and negative SEC scales in Study 2.**

|  | Min/Max | M | SD | Skewness | Kurtosis | Item total correlation if item deleted | α if item deleted |
|---|---|---|---|---|---|---|---|
| Positive SEC |  |  |  |  |  |  |  |
| PSEC1 | 1/5 | 4.02 | .75 | -.66 | .81 | .59 | .81 |
| PSEC2 | 2/5 | 3.77 | .76 | -.27 | -.21 | .73 | .74 |
| PSEC3 | 2/5 | 3.52 | .77 | -.23 | -.37 | .59 | .81 |
| PSEC4 | 1/5 | 3.62 | .80 | -.39 | -.07 | .70 | .76 |
| Negative SEC |  |  |  |  |  |  |  |
| NSEC1 | 1/5 | 2.78 | 1.02 | .06 | -.51 | .63 | .79 |
| NSEC2 | 1/5 | 2.75 | .96 | .32 | -.37 | .57 | .81 |
| NSEC3 | 1/5 | 2.99 | 1.04 | .07 | -.51 | .64 | .78 |
| NSEC4 | 1/5 | 2.91 | .93 | .05 | -.17 | .75 | .73 |

*Note.* The response scale of all items ranged from 1–5 and, as tested using Shapiro-Wilks tests, the collected data showed deviations from normality for all items.

Table 5 shows descriptive statistics of all scales reported in this study, including Positive and Negative SEC. Internal consistencies of our SEC scales were examined by calculating Cronbach's alpha and McDonald's omega indices showing good reliability for Positive SEC ($\alpha$ = .83; CI 95% [.79, .86] and $\omega$ = .83; CI 95% [.80, .86]) as well as Negative SEC ($\alpha$ = .82; CI 95% [.79, .86] and $\omega$ = .83; CI 95% [.79, .86]), according to conventions ([85]). The two factors Positive SEC and Negative SEC showed a small positive zero-order correlation between each other ($r$ = .17; $p$ < .001; CI 95% [.04, .29]) and a small to medium-sized positive correlation ($r$ = .27; $p$ < .001; CI 95% [.15, .39]), when controlling for social desirability, respectively; indicating that individuals reporting higher levels of Positive SEC also tended to report higher levels of Negative SEC.

**Internal validity.** The model fit for the two-factor model (Positive SEC and Negative SEC) was very good (CFI = .982; TLI = .973; SRMR = .045; RMSEA = .054 [.006, .088]) while the model fit of the single-factor model was unacceptable (CFI = .520; TLI = .327; SRMR = .189; RMSEA = .267 [.242, .293]). The superiority of the two-factor model over the single-factor model was further supported by a lower Akaike Information Criterion (AIC) value for the two-factor model (AIC = 4376.1) as compared to the single-factor model (AIC = 4712.9) as well as a highly significant model comparison ($\Delta\chi^2(1)$ = 1238.8, p < .001). As expected, all items showed high factor loadings with their corresponding factor, with positive standardized coefficients ranging from $\lambda$ = .63 to $\lambda$ = .88 (see Table 2).

**External validity.** To explore the external validity of both SEC scales, we obtained correlations of Positive SEC and Negative SEC with all criterion variables included in the study, controlling for social desirability (see Table 6).

Correlations with Measures of Empathy: While Positive SEC only showed small to medium-sized correlations with the assessed measures of empathy, Negative SEC showed a strong positive correlation with the scale AMES Affective Empathy ($r$ = .69) and a small positive correlation with the IRI Fantasy scale ($r$ = .21), but no correlations were found with the scales IRI Perspective Taking and AMES Cognitive Empathy.

Correlations with Measures of Positive/Negative Emotionality and Distress: Positive SEC was positively correlated with the scale BFI2 Activity/Energy Level ($r$ = .36), but uncorrelated with the other measures. Negative SEC showed a strong positive correlation with the scales IRI Personal Distress ($r$ = .58) and BFI2 Anxiety ($r$ = .52) and small to medium-sized correlations with the scales BFI2 Emotional Volatility ($r$ = .44), BFI2 Depressiveness ($r$ = .39), and PANAS Negative Affect ($r$ = .27); but no significant relations with the scales PANAS Positive Affect ($r$ = -.19) and BFI2 Activity/Energy Level ($r$ = -.12).

**Table 5. Descriptive statistics for all measures used in Study 2.**

| | Possible range | Min/ Max | M | SD | Cronbach's alpha | McDonald's omega total | Average inter item correlation | Skewness | Kurtosis | Normality (Shapiro-Wilk) |
|---|---|---|---|---|---|---|---|---|---|---|
| **Positive and Negative SEC Scales** | | | | | | | | | | |
| Positive SEC | 1–5 | 2/5 | 3.73 | .62 | .83 | .83 | .55 | -.31 | -.19 | NO |
| Negative SEC | 1–5 | 1/5 | 2.86 | .80 | .82 | .83 | .54 | .19 | .16 | NO |
| **Measures of Social Desirability** | | | | | | | | | | |
| Positive Qualities | 0–4 | 1.67/5 | 3.51 | .60 | .62 | .64 | .35 | -.19 | .03 | NO |
| Negative Qualities | 0–4 | 1/5 | 2.22 | .88 | .70 | .72 | .44 | .72 | -.05 | NO |
| **Measures of Empathy** | | | | | | | | | | |
| IRI Perspective Taking | 1–5 | 1.50/5 | 3.61 | .63 | .76 | .81 | .44 | -.19 | .12 | NO |
| IRI Fantasy | 1–5 | 1/5 | 3.37 | .73 | .75 | .81 | .44 | -.04 | .12 | NO |
| AMES Affective Empathy | 1–5 | 1/5 | 2.81 | .71 | .81 | .84 | .52 | .07 | .16 | NO |
| AMES Cognitive Empathy | 1–5 | 1.75/5 | 3.61 | .61 | .82 | .86 | .53 | -.04 | .02 | NO |
| **Measures of Positive/Negative Emotionality and Distress** | | | | | | | | | | |
| PANAS Positive Affect | 1–5 | 1.60/5 | 3.38 | .62 | .88 | .91 | .41 | -.26 | .15 | YES |
| PANAS Negative Affect | 1–5 | 1/4.10 | 1.77 | .65 | .89 | .92 | .46 | .73 | -.20 | NO |
| IRI Personal Distress | 1–5 | 1/4.75 | 2.68 | .71 | .76 | .80 | .44 | .14 | -.32 | NO |
| BFI2 Emotional Volatility | 1–5 | 1/4.75 | 2.70 | .73 | .75 | .80 | .43 | -.01 | -.19 | NO |
| BFI2 Depressiveness | 1–5 | 1/5 | 2.66 | .85 | .83 | .87 | .54 | .31 | -.56 | NO |
| BFI2 Anxiety | 1–5 | 1/5 | 2.99 | .72 | .69 | .77 | .36 | .02 | -.13 | NO |
| BFI2 Activity/ Energy Level | 1–5 | 1/5 | 3.17 | .78 | .73 | .79 | .41 | -.23 | -.13 | NO |
| **Measures of Well-Being and Mental/Physical Health** | | | | | | | | | | |
| EDS Depressiveness | 0–3 | 1/3.60 | 1.91 | .60 | .87 | .90 | .40 | .34 | -.74 | NO |
| GAD7 Anxiety | 0–3 | 1/3.86 | 1.87 | .62 | .88 | .91 | .50 | .87 | .50 | NO |
| PSS Stress | 1–5 | 1/4.50 | 2.69 | .69 | .87 | .90 | .39 | .21 | -.39 | NO |
| CHIPS Physical Symptoms | 1–5 | 1/4.50 | 2.47 | .80 | .80 | .86 | .33 | .32 | -.50 | NO |
| Life Satisfaction | 1–7 | 1/7 | 4.24 | 1.34 | .93 | .94 | .72 | -.37 | -.64 | NO |
| **Measures of Social/Interpersonal Functioning** | | | | | | | | | | |
| AMES Sympathy | 1–5 | 2.50/5 | 3.95 | .62 | .68 | .77 | .36 | -.09 | -.71 | NO |
| IRI Empathic Concern | 1–5 | 1.75/5 | 3.53 | .61 | .69 | .71 | .36 | .01 | -.26 | NO |
| BFI2 Sociability | 1–5 | 1/5 | 2.93 | .83 | .80 | .82 | .49 | -.10 | -.54 | NO |
| BFI2 Assertiveness | 1–5 | 1/5 | 3.14 | .74 | .78 | .80 | .47 | -.15 | -.08 | NO |
| BFI2 Compassion | 1–5 | 2.5/5 | 3.83 | .60 | .64 | .71 | .32 | -.17 | -.79 | NO |
| BFI2 Trust | 1–5 | 1.5/4.75 | 3.06 | .67 | .61 | .73 | .28 | -.00 | -.34 | NO |
| BFI2 Respectfulness | 1–5 | 2/5 | 3.97 | .61 | .73 | .80 | .42 | -.37 | -.17 | NO |

Correlations with Measures of Well-Being and Mental/Physical Health: Negative SEC showed significant medium-sized positive correlations with the scales EDS Depressiveness ($r$ = .29), GAD7 Anxiety ($r$ = .43), PSS Stress ($r$ = .36), and CHIPS Physical Symptoms ($r$ = .36) as well as a small negative correlation with the scale SWLS Life Satisfaction ($r$ = -.26). In contrast, Positive SEC was uncorrelated with all of these measures.

**Table 6. Partial correlations of positive and negative SEC with all criterion-variables while controlling for social desirability response tendencies.**

| | Positive SEC | | Negative SEC | |
|---|---|---|---|---|
| | _r_ | **95% CIs** | _r_ | **95% CI** |
| Measures of Empathy | | | | |
| IRI Perspective Taking | .20 ($p$ = .0021) | [.07, .31] | .04 ($p$ = .5065) | [-.08, .17] |
| IRI Fantasy | .30 ($p$ < .0001) | [.18, .41] | .21 ($p$ < .001) | [.09, .33] |
| AMES Affective Empathy | .28 ($p$ < .0001) | [.16, .39] | .69 ($p$ < .0001) | [.62, .75] |
| AMES Cognitive Empathy | .30 ($p$ < .0001) | [.19, .41] | .14 ($p$ = .0346) | [.01, .26] |
| Measures of Positive/Negative Emotionality and Distress | | | | |
| PANAS Positive Affect | .16 ($p$ = .0125) | [.04, .28] | -.19 ($p$ = .0027) | [-.31, -.07] |
| PANAS Negative Affect | -.02 ($p$ = .8104) | [-.14, .11] | .27 ($p$ < .0001) | [.15, .38] |
| IRI Personal Distress | .11 ($p$ = .0678) | [-.01, .24] | .58 ($p$ < .0001) | [.49, .66] |
| BFI2 Emotional Volatility | .02 ($p$ = .7149) | [-.10, .15] | .44 ($p$ < .0001) | [.33, .53] |
| BFI2 Depressiveness | -.03 ($p$ = .6127) | [-.16, .09] | .39 ($p$ < .0001) | [.28, .49] |
| BFI2 Anxiety | .09 ($p$ = .1636) | [-.04, .21] | .52 ($p$ < .0001) | [.42, .60] |
| BFI2 Activity/Energy Level | .36 ($p$ < .0001) | [.25, .46] | -.12 ($p$ = .0725) | [-.24, .01] |
| Measures of Well-being and Mental/Physical Health | | | | |
| EDS Depressiveness | .00 ($p$ = .9614) | [-.12, .13] | .29 ($p$ < .0001) | [.17, .40] |
| GAD7 Anxiety | .15 ($p$ = .0227) | [.02, .27] | .43 ($p$ < .0001) | [.33, .53] |
| PSS Stress | .04 ($p$ = .5348) | [-.09, .16] | .36 ($p$ < .0001) | [.24, .46] |
| CHIPS Physical Symptoms | .10 ($p$ = .1247) | [-.03, .22] | .36 ($p$ < .0001) | [.24, .46] |
| SWLS Life Satisfaction | .12 ($p$ = .0623) | [-.01, .24] | -.26 ($p$ < .001) | [-.37, -.13] |
| Measures of Social/Interpersonal Functioning | | | | |
| AMES Sympathy | .35 ($p$ < .0001) | [.23, .45] | .25 ($p$ < .001) | [.13, .35] |
| IRI Empathic Concern | .42 ($p$ < .0001) | [.31, .52] | .29 ($p$ < .0001) | [.18, .41] |
| BFI2 Sociability | .34 ($p$ < .0001) | [.23, .45] | -.14 ($p$ = .0288) | [-.26, -.02] |
| BFI2 Assertiveness | .15 ($p$ = .0180) | [.03, .27] | -.32 ($p$ < .0001) | [-.42, -.20] |
| BFI2 Compassion | .32 ($p$ < .0001) | [.21, .43] | .04 ($p$ = .4977) | [-.08, .17] |
| BFI2 Trust | .23 ($p$ < .001) | [.11, .35] | -.10 ($p$ = .1375) | [-.22, .03] |
| BFI2 Respectfulness | .17 ($p$ = .0077) | [.05, .29] | -.17 ($p$ = .0067) | [-.29, -.05] |

Correlations with Measures of Social/Interpersonal Functioning: Positive SEC showed medium-sized positive correlations with the scales AMES Sympathy ($r$ = .35), IRI Empathic Concern ($r$ = .42), BFI2 Sociability ($r$ = .34), and BFI2 Compassion ($r$ = .32). Negative SEC, in turn, was only weakly or uncorrelated with these measures, with only a few exceptions of small- to medium-sized correlations (i.e., BFI2 Assertiveness, IRI Empathic Concern, and AMES Sympathy).

## Discussion

First, regarding correlations with self-report measures of empathy (in terms of convergent/divergent validity), we did not find any substantial correlations between Negative SEC and most empathy scales. The only exception was a strong positive correlation with the AMES Affective Empathy scale. However, this positive association supports the convergent validity of our Negative SEC scale, given that the AMES Affective Empathy scale shows substantial overlap with our definition of SEC as well as with the wording of our Negative SEC items. In

comparison, Positive SEC showed only small to medium correlations with both cognitive and affective empathy components (not all of them significant). Overall, these findings suggest reasonable divergent validity in terms of both SEC scales' relations to self-reported empathy.

Second, regarding correlations with self-report measures of positive/negative emotionality and distress, our findings suggest asymmetric correlational associations with Positive SEC vs. Negative SEC. This asymmetric pattern implies that Negative SEC is related to an increased experience of emotional instability, distress, and negative emotions, including depressiveness and anxiety, while Positive SEC does not seem to be related to these measures. There were only few exceptions to this pattern: An individual's level of activity/energy as measured by the BFI2 scale was positively related to Positive SEC and, rather unexpectedly, both SEC scales were uncorrelated with positive affect as measured using the PANAS Positive Affect scale.

Third, this asymmetrical pattern was also evident regarding self-report measures of well-being and mental/physical health. Negative SEC was positively related to current experiences of depressiveness, anxiety, stress, and physical health and negatively related to their subjective well-being. Positive SEC, on the other hand, wasn´t correlated with any these variables.

Fourth, regarding measures of social/interpersonal functioning, our findings showed the opposite asymmetrical pattern. While Negative SEC did not seem to be substantially related to most of these variables, Positive SEC was related to greater interpersonal functioning and prosocial behavioral tendencies as measured by the scales AMES Sympathy, IRI Empathic Concern, BFI2 Sociability, and BFI2 Compassion. Again, there were a few exceptions from this asymmetric pattern: Assertiveness was negatively related to Negative SEC and uncorrelated with Positive SEC and Respectfulness was uncorrelated with both SEC scales. However, being assertive as measured by the BFI2 Assertiveness items reflects assuming control and influencing others and being respectful as measured by the BFI2 Respectfulness items reflects being polite and obliging. In that respect, these two scales differ from the other scales and possibly represent less prosocial facets of interpersonal interaction.

## General discussion

In the present work, we first aimed to investigate our novel scales' internal validity and reliability in terms of their psychometric properties, internal consistencies, and factor structure. Second, regarding the scales' external validity, we aimed to explore diverging patterns in the relations of Positive and Negative SEC with relevant criterion variables. To this end, we conducted two studies to examine our scales' internal validity using EFA (Study 1) and CFA (Study 2) and their external validity using correlations with measures of empathy, emotional experiences, mental health problems, and indicators of social functioning.

### Internal validity and reliability

Regarding the measure's internal validity, EFA in Study 1 revealed a two-factor structure with Positive SEC and Negative SEC as clearly distinct yet correlated factors. The factor structure was confirmed in Study 2 using CFA with the two-factor model showing a good model fit; clearly superior to a single-factor model as demonstrated by a chi-square difference test. In both studies, the two scales of our SEC Scale demonstrated good reliability in terms of Cronbach's alphas and McDonald's omegas, ranging between .76 and .83. Hence, the two newly developed SEC measures appear to be an internally valid and reliable self-report measures.

### External validity

Regarding the scales' external validity, we explored their relations to (1) self-report measures of empathy, (2) of positive/negative emotionality and distress, (3) of well-being and mental/

physical health, and (4) of social/interpersonal functioning. Overall, while both SEC scales seemed to show reasonable convergent and divergent validity based on their relations with the assessed empathy measures, we found diverging correlational patterns of Positive SEC vs. Negative SEC with the other criterion variables. On the one hand, Negative SEC was related to increased trait-like negative emotional experiences (BFI2 Depressiveness), emotionality (BFI2 Emotional Volatility), distress (IRI Personal Distress), and greater current mental health problems (EDS Depressiveness, GAD7 Anxiety, PSS Stress, SWLS Life Satisfaction) and physical symptoms (CHIPS Physical Symptoms). In contrast, Positive SEC appeared to be unrelated to these measures. On the other hand, Positive SEC was related to increased indicators of interpersonal functioning and prosocial tendencies, and other-oriented behaviors and cognitions (AMES Sympathy, IRI Empathic Concern, BFI2 Sociability, BFI2 Compassion). In turn, Negative SEC appeared to be unrelated to these variables.

Taken together, while previous research had provided initial scattered evidence of a certain degree of asymmetry in the pattern of relations of Positive SEC and Negative SEC with external variables–which is why we proposed to differentiate more clearly between the two dimensions in the first place–the pattern of relationships we found in our studies corroborated this divergence using two separate SEC scales. We conclude that future research will profit from differentiating these two dimensions of SEC, and explicitly considering the valence of emotions being transmitted form one person to another in the context of research on emotional contagion as a process and SEC as an individual's disposition.

## Limitations

While this work bears important new insights in the context of research on emotional contagion and SEC, it also bears some limitations which need to be taken into consideration when interpreting its findings. First, in both studies, we used a cross-sectional research design to assess the relations of Positive and Negative SEC to selected criterion variables. Therefore, it is important to be aware that our analyses cannot determine any temporal or causal relationship between the assessed variables. Second, we solely used self-report measures to assess SEC and all other variables. Thus, the reported results could be biased due to common method variance [112]. Third, while research on the process of emotional contagion in human social interaction has seemingly been increasing over the past years, SEC as an individual's disposition has been investigated less frequently in past research. As a result, some of the cited publications in the present work date back more than 10 years. Thus, more current scientific inquiries into SEC and its associations with relevant outcomes are needed, such as the present studies. Last, while both samples were sufficiently large in terms of statistical power for detecting medium and large effects as well as stability of correlations [113], the generalizability of the findings could still be limited due to our chosen sampling approaches and the fact that recruitment and data collection happened before the COVID-19 pandemic. In Study 1, we used a convenience sample of voluntarily recruited teachers. Therefore, it is possible that only highly motivated and healthy individuals took part in the study. In Study 2, we used an online sample recruited through a crowd-based online platform, also voluntarily but with a monetary incentive. While such crowd-based online samples have been found to be reasonably representative of the general population and to provide high-quality data [114], participants in these samples have been found to report increased levels of depressiveness with up to three times higher levels as compared to prevalence estimations in the general population [115]. Moreover, all data had been collected prior to the onset of the COVID-19 pandemic, which had been shown to have impacted individuals' well-being and mental health in the general population worldwide ([116]) and especially in highly vulnerable subgroups (e.g., [117,118]). Finally, both samples

were relatively homogeneous regarding cultural background and educational levels, and the language of all measures in our studies was German. Taken together, more diverse samples, recruited especially during or after the pandemic, are needed to replicate and corroborate our findings, and an English version of our novel self-report measure should be developed and validated in English-speaking populations.

### Directions for future research

First, to sustainably overcome the confusion regarding definitions and operationalizations of emotional contagion and SEC in future research and to foster a more consistent use of terms and concepts, we suggest—in line with previous theorists (e.g., [24]—that researchers explicitly define and transparently operationalize their view on emotional contagion and/or SEC in future studies.

Second, in addition to replicating our findings, more studies are needed to investigate the temporal stability and predictive validity of individuals' Positive SEC and Negative SEC. More specifically, their relations to other important variables, such as emotion regulation capacities, attachment security/insecurity, mentalization capacities, relationship quality, or prosocial behavior, should be investigated further.

Third, to go beyond self-report data in future studies, we suggest including other measures of emotional experiences, personality traits, and socially interactive behavior to cross-validate our new self-report measure, such as experience sampling [119,120], smartphone-based mobile sensing [121,122], or systematic behavioral coding. In that direction, the Facial Action Coding System (FACS; [123]) has proven to be a particularly suitable tool to quantify human facial expressions which represent a highly relevant channel to nonverbally communicate emotions to other individuals in socially interactive situations. Subsequently, the interpersonal transmission of facially expressed emotions could be investigated using nonlinear time series analyses, such as Cross-Recurrence Quantification [124]. Such fine-grained and dynamic methods are promising approaches to further validate our balanced Positive and Negative SEC scales as a measure of an individual's disposition to catch either positive and/or negative emotions when interacting with others.

### Supporting information

**S1 Appendix. Systematic review of published self-report measures of susceptibility to emotional contagion.** This review strives to (1) give a systematic overview of existing self-report instruments to measure individuals' susceptibility to emotional contagion (SEC), (2) to describe the theoretical framework of existing scales, and (3) to review the specific items that are used to assess individuals' SEC in different scales.
(DOCX)

**S1 Table. German items and related items in existing scales.**
(DOCX)

**S2 Table. Factor matrix of all 10 items initially tested in Study 1.**
(DOCX)

**S3 Table. Bivariate correlations of all measures with social desirability.**
(DOCX)

**S4 Table. Zero-order correlations of Positive and Negative SEC with all measures.**
(DOCX)

**S5 Table. Non-parametric Spearman correlations of Positive and Negative SEC with all measures.**
(DOCX)

## Author Contributions

**Conceptualization:** Anton K. G. Marx, Anne C. Frenzel.

**Data curation:** Anton K. G. Marx.

**Formal analysis:** Anton K. G. Marx.

**Investigation:** Anton K. G. Marx, Anne C. Frenzel.

**Methodology:** Anton K. G. Marx, Anne C. Frenzel, Daniel Fiedler.

**Project administration:** Anton K. G. Marx.

**Resources:** Anne C. Frenzel, Corinna Reck.

**Validation:** Anton K. G. Marx, Daniel Fiedler.

**Writing – original draft:** Anton K. G. Marx.

**Writing – review & editing:** Anton K. G. Marx, Anne C. Frenzel, Daniel Fiedler, Corinna Reck.

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
