## [Decision Letter · Decision Letter 0]

9 Aug 2023

PONE-D-23-10800Susceptibility to positive versus negative emotional contagion: First evidence on their distinction using a balanced self-report measurePLOS ONE

Dear Dr. Marx,

Thank you for submitting your manuscript to PLOS ONE. After careful consideration, we feel that it has merit but does not fully meet PLOS ONE’s publication criteria as it currently stands. Therefore, we invite you to submit a revised version of the manuscript that addresses the points raised during the review process.

We look forward to receiving your revised manuscript.

Kind regards,

Stefania Mancone

Academic Editor

PLOS ONE

Journal Requirements:

2. We note that you have referenced (ie. Bewick et al. [5]) which has currently not yet been accepted for publication. Please remove this from your References and amend this to state in the body of your manuscript: (ie “Bewick et al. [Unpublished]”) as detailed online in our guide for authors

Additional Editor Comments:

The journal Plos One publishes only high-quality work. Therefore, I ask the authors to review the manuscript, following the precise and timely directions of the reviewers. In anticipation, I wish you good work

Reviewers' comments:

Reviewer's Responses to Questions

**Comments to the Author**

1. Is the manuscript technically sound, and do the data support the conclusions?

Reviewer #1: Yes

Reviewer #2: Yes

2. Has the statistical analysis been performed appropriately and rigorously? 

Reviewer #1: Yes

Reviewer #2: Yes

3. Have the authors made all data underlying the findings in their manuscript fully available?

Reviewer #1: Yes

Reviewer #2: Yes

4. Is the manuscript presented in an intelligible fashion and written in standard English?

Reviewer #1: Yes

Reviewer #2: Yes

5. Review Comments to the Author

Reviewer #1: The reviewed article has a topic that is relevant and of interest to the public. A minor concern is due to the fact that the sample was recruited 5 years ago (2018), before the pandemic, but it is a consistent sample. The referencing used mostly dates back more than 10 years, which could perhaps be a criticism in the article's discussions.

Reviewer #2: Dear Editor,

Thank you for giving me the opportunity to review the article entitled “Susceptibility to positive versus negative emotional contagion: First evidence on their distinction using a balanced self-report measure”. I appreciated the clarity and accuracy of the article; nevertheless, I have some concerns that I listed below.

INTRODUCTION

The introduction is rich and well written. Below, some issues:

1) The definition and conceptualization of the EC seem to me a little too vague and inconsistent. Authors should clarify this point, according to the literature. Moreover, they should better clarify the distinction between EC and empathy.

2) The Authors should better discuss the relationship between positive and negative affect. This relationship can vary widely across samples, ranging from near zero (as reported) to strongly negative. Authors should discuss the existing evidence.

3) The Authors should better explain why they don’t expect an overarching construct.

4) The Authors presented an interesting systematic literature review of existing scales that explicitly addressed SEC (S1 Appendix); however, I think that Authors should clarify in the Introduction that there are other instruments which contains item of SEC (Positive and Negative Emotions). The Authors should also specify whether the 7 measures of the individuals’ SES targeted adults, or other age groups.

5) Why didn’t the Authors target also neutral contagion?

6) The Authors should better clarify why in the literature there is a focus on SEC of negative emotions.

METHOD

1) In the S2 Appendix, the Authors presented the items of their scale. It is unclear to me whether item PSEC1, PSEC4, NSEC4 have been newly developed by Authors. Why did they develop a new item for SEC for Negative Emotions? If I understood well, there are already existing validated tools to assess SEC for Negative Emotions. For adapted items, Authors should clarify the translation process. In which way, did the Authors assess content validity?

2) What does it mean that the participants have obtained German A levels? The Authors should clarify for a non-German audience.

3) It is not clear to me the rate of nonrespondents for Study 1 and Study 2.

4) The Authors should give more information about recruitment in Study 1. Was it part of a larger study? How many schools were involved? How did the Authors contact schools?

5) Cronbach’s alpha and McDonald’s omega for the scale of positive qualities of social desirability are low. Please clarify.

6. PLOS authors have the option to publish the peer review history of their article (what does this mean?). If published, this will include your full peer review and any attached files.

Reviewer #1: No

Reviewer #2: No

---

## [Author Response · Author response to Decision Letter 0]

25 Oct 2023

Response to Reviewers (Manuscript: PONE-D-23-10800)

-

JOURNAL REQUIREMENTS

Author Response: Thank you for pointing our attention to this issue. We ensured that the uploaded files of our resubmission followed those style requirements.

2. We note that you have referenced (ie. Bewick et al. [5]) which has currently not yet been accepted for publication. Please remove this from your References and amend this to state in the body of your manuscript: (ie “Bewick et al. [Unpublished]”) as detailed online in our guide for authors http://journals.plos.org/plosone/s/submission-guidelines#loc-reference-style

Author Response: Thank you for bringing this to our attention. However, we are not entirely sure about this feedback. In our manuscript, Bewick et al. was not cited, and our reference [5] was Hatfield et al. (1993). Nevertheless, we checked whether we had cited any not-yet published references and ensured that all our cited references were published and cited correctly in our manuscript.

-

ADDITIONAL EDITOR COMMENTS

The journal Plos One publishes only high-quality work. Therefore, I ask the authors to review the manuscript, following the precise and timely directions of the reviewers. In anticipation, I wish you good work.

Author Response: Thank you for allowing us to revise and resubmit our manuscript. Below, we describe in detail how we responded to each of the reviewers' comments.

-

REVIEWER #1

The reviewed article has a topic that is relevant and of interest to the public. A minor concern is due to the fact that the sample was recruited 5 years ago (2018), before the pandemic, but it is a consistent sample. The referencing used mostly dates back more than 10 years, which could perhaps be a criticism in the article's discussions.

Author Response: Thank you for your positive evaluation of our manuscript and for raising these important points. Regarding the sample recruitment date, we added a corresponding section in the discussion section including additional references, which reads as follows:

 Last, while both samples were sufficiently large in terms of statistical power for detecting medium and large effects as well as stability of correlations [111], the generalizability of the findings could still be limited due to our chosen sampling approaches and the fact that recruitment and data collection happened before the COVID-19 pandemic. In Study 1, we used a convenience sample of voluntarily recruited teachers. Therefore, it is possible that only highly motivated and healthy individuals took part in the study. In Study 2, we used an online sample recruited through a crowd-based online platform, also voluntarily but with a monetary incentive. While such crowd-based online samples have been found to be reasonably representative of the general population and to provide high-quality data [112], participants in these samples have been found to report increased levels of depressiveness with up to three times higher levels as compared to prevalence estimations in the general population [113]. Moreover, all data had been collected prior to the onset of the COVID-19 pandemic, which had been shown to have impacted individuals' well-being and mental health in the general population worldwide ([114]) and especially in highly vulnerable subgroups (e.g., [115,116]). Finally, both samples were relatively homogeneous regarding cultural background and educational levels, and the language of all measures in our studies was German. Taken together, more diverse samples, recruited especially during or after the pandemic, are needed to replicate and corroborate our findings, and an English version of our novel self-report measure should be developed and validated in English-speaking populations.

Author Response: Regarding the references dating back more than 10 years, we have added the following text to the discussion:

 Third, while research on the process of emotional contagion in human social interaction has seemingly been increasing over the past years, SEC as an individual's disposition has been investigated less frequently in past research. As a result, some of the cited publications in the present work date back more than 10 years. Thus, more current scientific inquiries into SEC and its associations with relevant outcomes are needed, such as the present studies.

-

REVIEWER #2

Thank you for giving me the opportunity to review the article entitled “Susceptibility to positive versus negative emotional contagion: First evidence on their distinction using a balanced self-report measure”. I appreciated the clarity and accuracy of the article; nevertheless, I have some concerns that I listed below.

Author Response: Thank you for your generally positive evaluation of our manuscript and for raising a number of very important and constructive comments that helped us to revise the paper. Below, we outline how we responded to each of your comments.

INTRODUCTION

The introduction is rich and well written. Below, some issues:

1) The definition and conceptualization of the EC seem to me a little too vague and inconsistent. Authors should clarify this point, according to the literature. Moreover, they should better clarify the distinction between EC and empathy.

Author Response: Thank you for pointing this out. We agree that it is important to clarify the definition of emotional contagion and its distinction from the concept of empathy, and we also observed that the various definitions found in the existing literature often seem to be overlapping or vague. While revising this manuscript, we made an effort to provide as precise as possible definitions while recognizing the variety of perspectives in the existing literature. The revised section read as follows: 

 The idea of emotional contagion has been around since the 18th century [12,13], and it was introduced to scientific fields related to psychology at least 100 years ago [14-17]. Since then, various terms have been used to describe this process, including contagion, transmission, transfer, crossover, resonance, or mirroring [3,7,18-23], and different underlying mechanisms have been proposed [4]. While previous studies have often been unclear and inconsistent in their conceptualizations and definitions [24], emotional contagion can be defined - in agreement with several theorists - as a basic interpersonal process that comprises the mostly automatic and unintentional transmission of emotions from an individual to one or more other individuals resulting in a more or less congruent emotional experience [2,5,7,8]. 

 In that sense, it can be delineated clearly from the much broader concept of empathy, which itself has recently been discussed in recent reviews as a rather vague and manyfold umbrella term comprising a range of more precisely defined lower-level concepts (see 24-26], for critical discussions of empathy definitions). In line with these reviews, empathy definitions typically encompass understanding others’ emotions and taking their perspective (i.e., cognitive empathy), sharing someone else’s emotions and feeling prosocial concern, compassion, or sympathy for others (i.e., affective empathy), and sometimes even the behavioral tendencies and reactions resulting thereof as well as accurately recognizing others' emotional states [24-26]. However, this rather diverse collection of processes and capacities has been criticized and discussed as a threat to clear and consistent conceptualizations that are needed for scientific advancement and, instead, it has been suggested to bypass the term empathy in favor of more precise definitions of lower-level constructs, such as emotional contagion (e.g., 24,25). 

 Thus, in contrast to empathy, emotional contagion represents a purely affective response to other individuals' emotional experiences (instead of cognitive role-/perspective-taking), it is based on unintentional, automatic, and mostly unconscious processes (instead of deliberately imagining being in another person's situation), and it does not necessarily involve any behavioral reactions towards others (instead of comforting or helping another person). Most critically, though, emotional contagion does not require to distinguish between one's own emotions and the other person's emotions (instead of an at least minimum level of self-other distinction or awareness of another individual's situation that is postulated as a defining feature of empathy) [24-30]. In other words, emotional contagion means that another individual's emotions become my own emotions, while empathy means understanding and responding to another individual's emotional experiences while distinguishing between my response and the other person's emotions or situation. 

Author Response: Additionally, to foster a more precise and transparent use of definitions of emotional contagion and SEC, we added the following paragraph in the Discussion section under "Directions for Future Research".

 First, to sustainably overcome the confusion regarding definitions and operationalizations of emotional contagion and SEC in future research and to foster a more consistent use of terms and concepts, we suggest - in line with previous theorists (e.g.,[24]) - that researchers explicitly define and transparently operationalize their view on emotional contagion and/or SEC in future studies.

2) The Authors should better discuss the relationship between positive and negative affect. This relationship can vary widely across samples, ranging from near zero (as reported) to strongly negative. Authors should discuss the existing evidence.

Author Responses: Thank you for bringing this to our attention. We agree that it is important to discuss the existing evidence on the association between positive and negative affect more thoroughly. To this end, we have revised the respective paragraphs as follows, bringing in more evidence from this body of literature:

 Originally, SEC has been conceptualized as a unidimensional and global construct. In the present work, we argue for a more complex perspective on an individual’s SEC. Theoretically, our reasoning is rooted in Watson and Tellegen’s [54] consensual model of positive and negative affect. In contrast to emotion theorists who argued for a specific set of discrete emotions [55-58], Watson and Tellegen proposed a more parsimonious model based on the reanalysis of several datasets with positive and negative affect as two general and relatively independent dimensions of emotional experiences that can be subdivided into specific discrete emotions reflecting the valence of the dimensions, such as happiness or anxiety [54,59-61]. 

 Since Watson and Tellegen’s proposal in 1985 [54], a range of empirical studies have provided evidence for this assumption. For example, in a study investigating college students' affect over several class sessions and how it is altered by the feedback on their performance, positive and negative affect were found to be unrelated in each measurement. Moreover, while success feedback increased positive affect, it did not impact students' negative affect, and vice versa ([62]). In addition, several studies investigated certain context factors influencing the relationship between positive and negative affect. For instance, in a study examining the influence of culture and gender on the relationship between positive and negative affect, different correlational patterns were found for the subgroups of American men (weak negative correlation between positive and negative emotions) and women (strong negative correlation) versus Chinese men (weak positive correlation) and women (strong positive correlation), respectively ([63]). Additionally, previous studies examining the influence of experienced stress on this association found that greater levels of stress lead to a stronger negative correlation between positive and negative affect ([64,65]). In studies investigating the relation between positive and negative affect for both state and trait measures, the two dimensions were found to be unrelated at the trait level but showed a significant small negative correlation when considering a shorter time interval ([66,67]). Taken together, the existing evidence suggests that people who frequently and intensively experience positive emotions do not systematically experience elevated or decreased levels of negative emotions. As such, the proposal of positive and negative affect as two separable and -more or less- independent dimensions provides a plausible and promising theoretical framework for our reasoning. 

 In line with this framework, we propose two distinct dimensions of SEC: The SEC of positive emotions (Positive SEC) and the SEC of negative emotions (Negative SEC).

3) The Authors should better explain why they don’t expect an overarching construct.

Author Responses: Thank you for raising this important point. In the present contribution, we propose two distinct dimensions of SEC (Positive and Negative SEC) reflecting the emotional valence of the emotions being transmitted via the process of emotional contagion. The key point of the present work, however, is not that we rule out any covariation between these two dimensions (Positive and Negative SEC) in terms of a potential overarching construct which could explain small amounts of variance. Instead, we propose that the conceptual separation of these two dimensions (even if correlated) is insightful and promising for future research as demonstrated by the diverging correlational patterns with other relevant variables. To clarify this, we revised the respective sentences as follows:

 In line with Watson and Tellegen’s model, we do not necessarily expect an antagonistic relation between Positive and Negative SEC (i.e., a higher Positive SEC does not necessarily lead to a lower Negative SEC). While we do not rule out any covariation between the two dimensions, we do not propose that there is a strong higher-order personal tendency to catch emotions of either valence from other people (resulting in a strong positive correlation between Positive and Negative SEC and explaining most of the variance). Instead, we expect Positive SEC and Negative SEC to be largely independent from one another and linked differentially to variables in several important domains of human functioning, namely emotional experiences, social interaction, and individuals’ mental and physical health. 

4) The Authors presented an interesting systematic literature review of existing scales that explicitly addressed SEC (S1 Appendix); however, I think that Authors should clarify in the Introduction that there are other instruments which contains item of SEC (Positive and Negative Emotions). The Authors should also specify whether the 7 measures of the individuals’ SES targeted adults, or other age groups.

Author Responses: Thank you for pointing this out. To clarify this point, we now mention that there are 21 scales focusing on different forms of empathy that contain items resembling emotional contagion and we added the information on the target groups of the scales that explicitly addressed SEC. We revised the respective paragraph which now reads as follows:

 Methodologically, previous research relied on self-report measures to assess an individual’s SEC. Following an extensive and systematic literature review (see supporting information S1 Appendix), we identified seven published scales that explicitly addressed SEC (e.g., the Emotional Contagion Scale [44] and the Emotional Empathic Tendency Scale [7]) and 21 scales focusing on different forms of empathy that contained items resembling emotional contagion. Of the seven scales addressing SEC explicitly, four targeted adults ([7,44-46]), one targeted adults and adolescents ([47]), one specifically targeted parents ([48]), and one targeted children ([49]), respectively. All of the seven instruments have been constructed based on t

---

## [Decision Letter · Decision Letter 1]

24 Mar 2024

PONE-D-23-10800R1Susceptibility to positive versus negative emotional contagion: First evidence on their distinction using a balanced self-report measurePLOS ONE

Dear Dr. Marx,

Thank you for submitting your manuscript to PLOS ONE. After careful consideration, we feel that it has merit but does not fully meet PLOS ONE’s publication criteria as it currently stands. Therefore, we invite you to submit a revised version of the manuscript that addresses the points raised during the review process.

Please address reviewer’s comments. Please submit your revised manuscript by May 08 2024 11:59PM. If you will need more time than this to complete your revisions, please reply to this message or contact the journal office at plosone@plos.org. Please include the following items when submitting your revised manuscript:A rebuttal letter that responds to each point raised by the academic editor and reviewer(s). You should upload this letter as a separate file labeled 'Response to Reviewers'.A marked-up copy of your manuscript that highlights changes made to the original version. You should upload this as a separate file labeled 'Revised Manuscript with Track Changes'.An unmarked version of your revised paper without tracked changes. You should upload this as a separate file labeled 'Manuscript'.If applicable, we recommend that you deposit your laboratory protocols in protocols.io to enhance the reproducibility of your results. Protocols.io assigns your protocol its own identifier (DOI) so that it can be cited independently in the future. For instructions see: https://journals.plos.org/plosone/s/submission-guidelines#loc-laboratory-protocols. Additionally, PLOS ONE offers an option for publishing peer-reviewed Lab Protocol articles, which describe protocols hosted on protocols.io. Read more information on sharing protocols at https://plos.org/protocols?utm_medium=editorial-email&utm_source=authorletters&utm_campaign=protocols.

We look forward to receiving your revised manuscript.

Kind regards,

Majed Sulaiman Alamri, PhD

Academic Editor

PLOS ONE

Journal Requirements:

Reviewers' comments:

Reviewer's Responses to Questions

**Comments to the Author**

1. If the authors have adequately addressed your comments raised in a previous round of review and you feel that this manuscript is now acceptable for publication, you may indicate that here to bypass the “Comments to the Author” section, enter your conflict of interest statement in the “Confidential to Editor” section, and submit your "Accept" recommendation.

Reviewer #2: All comments have been addressed

Reviewer #3: (No Response)

2. Is the manuscript technically sound, and do the data support the conclusions?

Reviewer #2: (No Response)

Reviewer #3: Yes

3. Has the statistical analysis been performed appropriately and rigorously? 

Reviewer #2: (No Response)

Reviewer #3: Yes

4. Have the authors made all data underlying the findings in their manuscript fully available?

Reviewer #2: (No Response)

Reviewer #3: Yes

5. Is the manuscript presented in an intelligible fashion and written in standard English?

Reviewer #2: (No Response)

Reviewer #3: Yes

6. Review Comments to the Author

Reviewer #2: (No Response)

Reviewer #3: The manuscript is solid and rigorous in its scientific content. The data has been well analyzed and the conclusions are clear and relevant. The sample size is appropriate for studies of this nature. However, I have some minor suggestions that I detail below:

1. The authors explain that the sample of Study 1 consisted of 257 teachers (line 241, page 12). A few lines below (lines 255 and 256), they mention that the questionnaires were sent to all teachers with a response rate of 53.2%. This could give the mistaken idea that the original sample was 257 teachers which was reduced by almost 50%. Were the questionnaires sent to a larger group, corresponding to a broader project (line 230) and only 257 participants responded? Clarifying this point in this section would be beneficial.

2. Regarding tables preparation, following common practice, it is suggested to avoid including columns that only contain constant values (e.g. “Possible range” and “Normality” in Tables 1 and 4). Such information could be simply mentioned in a table footnote.

3. On the other hand, it is expected that the explanations of the results precede table presentation. This criterion seems not to be met concerning Table 1. This shows 4 items for each scale, when it had previously been explained that the scales contained 5 items each. The reader may be confused until reaching the explanation offered below, on line 328. It would be good to correct this matter.

4. Something similar occurs with Table 2, which includes results from Study 2, not yet been introduced or explained in the text.

5. Concerning Study 2, the authors propose that the objective was to investigate the validity and reliability in terms of its psychometric properties, internal consistencies, and factor structure. However, this has already been tested in Study 1. I suggest slightly modifying the objective wording to reflect that the purpose of Study 2 was to “corroborate,” “confirm,” and/or “expand” the evidence of validity and reliability collected in Study 1.

6. It would also be recommended that the internal consistency values of the scales used to study the external validity of the SEC (empathy, positive/negative emotionality, etc.) be reported in the Measures section and not in the Results.

7. On the other hand, I think it might be useful to introduce somewhere in the manuscript, before presenting the desirability scale, a brief justification of why it was considered important to include it.

8. Finally, perhaps it could be interesting to review whether it is appropriate to use parametric correlations, depending on the compliance (or not) of the normality of the data.

This is all. I appreciate the opportunity to review this valuable article.

7. PLOS authors have the option to publish the peer review history of their article (what does this mean?). If published, this will include your full peer review and any attached files.

Reviewer #2: No

Reviewer #3: No

---

## [Author Response · Author response to Decision Letter 1]

27 Mar 2024

Response to Reviewers (Manuscript: PONE-D-23-10800R1)

Thank you for allowing us to revise and resubmit our manuscript again. Below, we describe in detail how we responded to the reviewer's comments.

Review Comments to the Author

REVIEWER #3

The manuscript is solid and rigorous in its scientific content. The data has been well analyzed and the conclusions are clear and relevant. The sample size is appropriate for studies of this nature. However, I have some minor suggestions that I detail below:

Author Response: Thank you for your positive evaluation of our manuscript and for raising these important points. Below, we outline how we responded to each of your comments.

1. The authors explain that the sample of Study 1 consisted of 257 teachers (line 241, page 12). A few lines below (lines 255 and 256), they mention that the questionnaires were sent to all teachers with a response rate of 53.2%. This could give the mistaken idea that the original sample was 257 teachers which was reduced by almost 50%. Were the questionnaires sent to a larger group, corresponding to a broader project (line 230) and only 257 participants responded? Clarifying this point in this section would be beneficial.

Author Response: Thank you for pointing this out. We agree that it is important to clarify the explanation of the response rate and the final sample size. The revised section reads as follows (in the "Materials and Methods" section of Study 1 under "Sample and Procedure"): 

 All participants were recruited voluntarily in 2017 through convenience sampling and direct contact with the schools or the teachers (i.e., schools and teachers were contacted via email, telephone, and written informational letters). Overall, more than 480 paper-pencil questionnaires were sent out, with a response rate of 53.2 % resulting in our final sample size and exceeding response rates of previous paper-pencil studies in this specific population [78,79]. No incentives were given for participation and no identifiers that could link individual participants to their results were obtained. Thus, all analyses were conducted on anonymous data. All participants gave their informed consent and the study was conducted in accordance with the ethical standards of the APA and the Declaration of Helsinki. The research project received a formal waiver of ethical approval by the ethics review board of the first author’s institution.

2. Regarding tables preparation, following common practice, it is suggested to avoid including columns that only contain constant values (e.g. “Possible range” and “Normality” in Tables 1 and 4). Such information could be simply mentioned in a table footnote.

Author Responses: Thank you for bringing this to our attention. We agree that the Tables 1 and 4 would benefit from omitting these two categories entailing constant values and mentioning the respective information in table notes. The table note that was added to Table 1 reads as follows: 

 Note. Following EFA, two items were excluded based on their factor loadings and/or cross-loadings resulting in 4 items per scale (see EFA results for detailed information). The response scale of all items ranged from 1-5 and, as tested using Shapiro-Wilks tests, the collected data showed deviations from normality for all items.

The table note that was added to Table 4 reads as follows: 

 Note. The response scale of all items ranged from 1-5 and, as tested using Shapiro-Wilks tests, the collected data showed deviations from normality for all items.

3. On the other hand, it is expected that the explanations of the results precede table presentation. This criterion seems not to be met concerning Table 1. This shows 4 items for each scale, when it had previously been explained that the scales contained 5 items each. The reader may be confused until reaching the explanation offered below, on line 328. It would be good to correct this matter.

Author Responses: Thank you for pointing this out. We added this important information in the table note that was added to Table 1. The revised note reads as follows:

 Note. Following EFA, two items were excluded based on their factor loadings and/or cross-loadings resulting in 4 items per scale (see EFA results for detailed information). The response scale of all items ranged from 1-5 and, as tested using Shapiro-Wilks tests, the collected data showed deviations from normality for all items.

4. Something similar occurs with Table 2, which includes results from Study 2, not yet been introduced or explained in the text.

Author Responses: Thank you for raising this point. We added the following short reference to Study 2 in the sentence in which Table 2 is mentioned for the first time (in the "Results and Discussion" section of Study 1 under "Exploratory Factor Analyses"):

 Based on item content, Factor 1 was labeled “Positive SEC” and Factor 2 was labeled “Negative SEC” (see Table 2 for the factor loadings of the final 8 items in Study 1 and Study 2).

Additionally, we added the following table note to Table 2 for further clarification:

 Note. In Study 1, EFA was conducted to test the assumption of two separate dimensions of SEC. In Study 2, CFA was conducted to replicate the factor structure that was found in Study 1. For both studies, only factor loading >.1 are displayed.

5. Concerning Study 2, the authors propose that the objective was to investigate the validity and reliability in terms of its psychometric properties, internal consistencies, and factor structure. However, this has already been tested in Study 1. I suggest slightly modifying the objective wording to reflect that the purpose of Study 2 was to “corroborate,” “confirm,” and/or “expand” the evidence of validity and reliability collected in Study 1.

Author Responses: Thank you for bringing this to our attention. We agree that a slight modification of the study's objectives would better reflect the purpose of Study 2. We revised the respective section as follows:

 In Study 2, we had two primary goals: First, we aimed to corroborate and expand the evidence on our novel scales' validity and reliability. 

6. It would also be recommended that the internal consistency values of the scales used to study the external validity of the SEC (empathy, positive/negative emotionality, etc.) be reported in the Measures section and not in the Results.

Author Responses: Thank you for bringing this to our attention. We moved the internal consistency values to the Measures section. Additionally, we decided to keep the internal consistency values in Table 5 as well, in order to make it easier for the reader to compare all psychometric properties for each scale and across all scales within the same table. The revised section reads as follows (in the "Materials and Methods" section of Study 2 under "Measures"):

Measures of Empathy: To assess different components of empathy, we used scales from the Interpersonal Reactivity Index (IRI; [86,95]) and the Adolescent Measure of Empathy and Sympathy (AMES; [87]), which has been validated in adult samples as well [96]. The IRI Perspective-Taking scale (Cronbach’s alpha α = .76; McDonald’s omega total ω = .81) measures an individual’s tendency to adopt another's perspective to find out what another person might be thinking, thus, representing the cognitive empathy component. The IRI Fantasy scale (α = .75; ω = .81) measures an individual’s tendency to imaginatively transpose oneself into the feelings and actions of fantasy characters. The AMES Affective Empathy scale (α = .81; ω = .84) aims to measure the tendency to vicariously experience another person’s emotions which is quite similar to the concept of emotional contagion. The AMES Cognitive Empathy scale (α = .82; ω = .86) aims to measure an individual’s capacity to cognitively understand another person’s emotions.

Measures of Positive/Negative Emotionality and Distress: To assess different aspects of individuals’ emotionality and distress, we used scales of the IRI and the Big-Five-Inventory 2 (BFI2; [88,97]) as well as the Positive and Negative Affect Schedule (PANAS; [59,98]). The IRI Personal Distress scale (α = .76; ω = .80) measures an individual’s self-oriented tendency to experience distress and anxiety in highly emotional interpersonal situations, such as emergencies. Within the BFI2, the Big Five personality traits can be further subdivided in more differentiated facets. From the Negative Emotionality trait (formerly Neuroticism), we used the BFI2 Emotional Volatility (α = .75; ω = .80), BFI2 Depressiveness (α = .83; ω = .87), and BFI2 Anxiety (α = .69; ω = .77) facets and the BFI2 Activity/Energy Level (α = .73; ω = .79) facet from the Extraversion trait, respectively. The PANAS scales (Positive Affect and Negative Affect) assess an individual’s general experience of positive (α = .88; ω = .91) and negative affect (α = .89; ω = .92) and consist of adjectives describing discrete emotional experiences that are either positive or negative in their valence (instruction: “How do you feel in general?”).

Measures of Well-being and Mental/Physical Health: We used the Edinburgh Depression Scale (EDS; [89,99,100]) to assess current depressive symptoms (α = .87; ω = .90), the General Anxiety Disorder 7 questionnaire (GAD-7; [90,101]) to assess current symptoms of general anxiety (α = .88; ω = .91), and the Perceived Stress Scale (PSS-10; [91,102]) to assess individuals’ current levels of stress (α = .87; ω = .90). To assess individuals’ current physical symptoms (α = .80; ω = .86), we used a short version of the Cohen-Hoberman-Inventory of Physical Symptoms (CHIPS; [92,103]) and to assess individuals’ satisfaction with their lives and current living conditions (α = .93; ω = .94), we used the Satisfaction with Life Scale (SWLS; [93]). 

Measures of Social/Interpersonal Functioning: To assess different aspects of individuals’ social and interpersonal functioning, we used scales of the IRI, the AMES, and specific facets of BFI2 personality traits. The AMES Sympathy subscale (α = .68; ω = .77) aims to measure an individual’s tendency to feel concerned or sorrow for another person. The IRI Empathic Concern subscale (α = .69; ω = .71) aims to measure an individual’s tendency to feel concern and sympathy towards others. From the personality trait extraversion, we used the BFI2 Sociability (α = .88; ω = .82) and BFI2 Assertiveness (α = .78; ω = .80) facets and the BFI2 Compassion (α = .64; ω = .71), BFI2 Trust (α = .61; ω = .73), and BFI2 Respectfulness (α = .73; ω = .80) facets from the personality trait Agreeableness, respectively. 

 Social Desirability: Given that self-reported health-related and clinical psychological variables have been reported to be associated with social desirability ([94]), we decided to assess the participants' social desirability response tendencies. To this end, we used a short German measure (KSE-G; [94]) of the tendency to either exaggerate one’s positive qualities (α = .62; ω = .64) or to conceal negative qualities (α = .70; ω = .72)

7. On the other hand, I think it might be useful to introduce somewhere in the manuscript, before presenting the desirability scale, a brief justification of why it was considered important to include it.

Author Responses: Thank you for raising this point. We added the following brief justification right before presenting the social desirability scale in the Measures section:

 Social Desirability: Given that self-reported health-related and clinical psychological variables have been reported to be associated with social desirability ([94]), we decided to assess the participants' social desirability response tendencies. To this end, we used a short German measure (KSE-G; [94]) of the tendency to either exaggerate one’s positive qualities (α = .62; ω = .64) or to conceal negative qualities (α = .70; ω = .72)

8. Finally, perhaps it could be interesting to review whether it is appropriate to use parametric correlations, depending on the compliance (or not) of the normality of the data.

Author Responses: Thank you for bringing up this important question. We agree that the choice of parametric correlations in this study should be explained more transparently. 

First, given that Shapiro-Wilk tests are vulnerable to sample size and tend to become significant more easily in larger samples, it is generally suggested to inspect the distribution of the collected data visually as well as their skewness levels in addition to applying a Shapiro-Wilk test (e.g., Field et al., 2012). 

Thus, we visually inspected our data using histogram plots. Of all 27 variables, only the variables PANAS negative affect and GAD7 anxiety appeared to be clearly skewed to the right (i.e., positively skewed) based on these plots. This impression was substantiated when looking at their skewness values reported in Table 5, whereas most other variables appeared to be approximately symmetric (with values of zero or close to zero) or only moderately skewed (with values between -.5 and +.5). 

Second, Pearson correlations have long been argued to be robust against violations of the normality assumption (e.g., Havlicek & Peterson, 1976), especially in larger samples (e.g., due to the central limit theorem). 

Taken together, we consider the distribution of our variables as sufficiently normal and Pearson correlations as sufficiently robust to use parametric correlations.

However, to increase research transparency, we have calculated non-parametric Spearman correlations with positive SEC and negative SEC for all variables and added these correlations as an additional supplement file (see supplement S6). 

Regarding positive SEC, for example, the correlation with empathic concern changed from .47 to .42 and remained highly significant (p < .0001). The correlation with EDS depressiveness changed from .00 to .-.04 but remained far from significant (i.e., p = .9614/.5009).

Regarding negative SEC, for instance, the correlation with the variable BFI2 Trust changed from -.10 to -.19 but did not reach significance (p = .1375/.0022). On the other hand, the correlation with the variable PANAS positive affect changed from -.19 to -.03 and remained far from statistical significance (p = .0027/.6366). 

In sum, while some correlations are slightly larger or smaller when applying non-parametric correlations, the asymmetric correlational patterns for positive SEC versus negative SEC as empirical foundation of our argument for distinct dimensions of SEC, appears to hold true. 

To further maximize research transparency, we have added the R-code for these histogram plots as well as the non-parametric Spearman correlations to the R-script "study-02.R" in the supplement and updated the file that is available in our OSF repository. 

To include these considerations in the manuscript, we have added the following explanation in (in the "Materials and Methods" section of Study 2 under "Statistical Analyses") as well as an additional Table in the supplement (S6) comprising the non-parametric correlations of Positive SEC and Negative SEC with all variables:

While the distribution of almost all variables were judged as non-normal based on Shapiro-Wilk tests, we still consider the distribution of our variables sufficiently normal based on visual inspection of the collected data (all data visualizations are included in the analysis scripts in the supplement) and the respective skewness levels ([85]), especially in light of the generally assumed robustness of Pearson correlations against violations of the normality assumption (e.g., [110]; see supporting information S6 Table for supplemental non-parametric Spearman correlations of Positive and Negative SEC with all measures.).

---

## [Decision Letter · Decision Letter 2]

16 Apr 2024

Susceptibility to positive versus negative emotional contagion: First evidence on their distinction using a balanced self-report measure

PONE-D-23-10800R2

Dear Dr. Marx,

We’re pleased to inform you that your manuscript has been judged scientifically suitable for publication and will be formally accepted for publication once it meets all outstanding technical requirements.

Kind regards,

Majed Sulaiman Alamri, PhD

Academic Editor

PLOS ONE

Additional Editor Comments (optional):

Reviewers' comments:

Reviewer's Responses to Questions

**Comments to the Author**

1. If the authors have adequately addressed your comments raised in a previous round of review and you feel that this manuscript is now acceptable for publication, you may indicate that here to bypass the “Comments to the Author” section, enter your conflict of interest statement in the “Confidential to Editor” section, and submit your "Accept" recommendation.

Reviewer #2: All comments have been addressed

Reviewer #3: All comments have been addressed

2. Is the manuscript technically sound, and do the data support the conclusions?

Reviewer #2: Yes

Reviewer #3: Yes

3. Has the statistical analysis been performed appropriately and rigorously? 

Reviewer #2: Yes

Reviewer #3: Yes

4. Have the authors made all data underlying the findings in their manuscript fully available?

Reviewer #2: Yes

Reviewer #3: Yes

5. Is the manuscript presented in an intelligible fashion and written in standard English?

Reviewer #2: Yes

Reviewer #3: Yes

6. Review Comments to the Author

Reviewer #2: I would like to thank the authors for their thorough revisions to the manuscript. It appears that they have effectively addressed all the raised issues. Consequently, I recommend accepting the paper.

Reviewer #3: The authors have adequately addressed my comments raised in a previous round of review and I feel that this manuscript is now acceptable for publication.

7. PLOS authors have the option to publish the peer review history of their article (what does this mean?). If published, this will include your full peer review and any attached files.

Reviewer #2: No

Reviewer #3: No

---

## [Editor Report · Acceptance letter]

26 Apr 2024

PONE-D-23-10800R2 

PLOS ONE

Dear Dr. Marx, 

I'm pleased to inform you that your manuscript has been deemed suitable for publication in PLOS ONE. Congratulations! Your manuscript is now being handed over to our production team.

Kind regards, 

on behalf of

Dr. Majed Sulaiman Alamri 

Academic Editor

PLOS ONE